# Ag-CuO-Decorated Ceramic Membranes for Effective Treatment of Oily Wastewater

**DOI:** 10.3390/membranes13020176

**Published:** 2023-02-01

**Authors:** Amos Avornyo, Arumugham Thanigaivelan, Rambabu Krishnamoorthy, Shadi W. Hassan, Fawzi Banat

**Affiliations:** 1Department of Chemical Engineering, Khalifa University, Abu Dhabi 127788, United Arab Emirates; 2Center for Membranes and Advanced Water Technology (CMAT), Khalifa University of Science and Technology, Abu Dhabi 127788, United Arab Emirates

**Keywords:** ceramic membranes, TiO_2_/ZrO_2_, membrane coating, nanocomposites, oil/water separation

## Abstract

Although ultrafiltration is a reliable method for separating oily wastewater, the process is limited by problems of low flux and membrane fouling. In this study, for the first time, commercial TiO_2_/ZrO_2_ ceramic membranes modified with silver-functionalized copper oxide (Ag-CuO) nanoparticles are reported for the improved separation performance of emulsified oil. Ag-CuO nanoparticles were synthesized via hydrothermal technique and dip-coated onto commercial membranes at varying concentrations (0.1, 0.5, and 1.0 wt.%). The prepared membranes were further examined to understand the improvements in oil-water separation due to Ag-CuO coating. All modified ceramic membranes exhibited higher hydrophilicity and decreased porosity. Additionally, the permeate flux, oil rejection, and antifouling performance of the Ag-CuO-coated membranes were more significantly improved than the pristine commercial membrane. The 0.5 wt.% modified membrane exhibited a 30% higher water flux (303.63 L m^−2^ h^−1^) and better oil rejection efficiency (97.8%) for oil/water separation among the modified membranes. After several separation cycles, the 0.5 wt.% Ag-CuO-modified membranes showed a constant permeate flux with an excellent oil rejection of >95% compared with the unmodified membrane. Moreover, the corrosion resistance of the coated membrane against acid, alkali, actual seawater, and oily wastewater was remarkable. Thus, the Ag-CuO-modified ceramic membranes are promising for oil separation applications due to their high flux, enhanced oil rejection, better antifouling characteristics, and good stability.

## 1. Introduction

The Middle East and North Africa (MENA) region is the most vulnerable to climate change and water-stressed in the world [1]. Crude oil is the region’s main export commodity, the extraction of which, unfortunately, puts further strain on already scarce water resources. It is estimated that crude oil extraction in the region uses approximately 2.73 × 10^9^ m^3^ of water annually [2], mainly being injected for drilling oil extraction and shoring up the dwindling pressures of aging oil fields [3]. Some of this injected water, together with connate water, ends up being co-produced with the oil. The average ratio between the volume of this produced water and the volume of produced oil is 3:1, making the produced water the largest waste stream in the petroleum industry [4]. Produced water, among other pollutants, contains dispersed oil droplets and causes severe environmental destruction when disposed of without treatment [5]. Successful treatment of this aqueous waste not only prevents probable environmental hazards but also provides more reusable water with oil recovery, thus reducing the dependence on freshwater sources for drilling and production of crude.

Most conventional treatment approaches (skimmers, solidifiers, dispersants, microbes, adsorbents, controlled combustion, etc.) for treating oily wastewater have practical limitations, especially in the separation of stable oil-in-water (o/w) emulsions (oil droplets size < 10 μm), thus unable to meet strict environmental requirements for the discharge of oily effluents [6,7,8]. Recently, membrane separation, especially ultrafiltration (UF), has been considered a more effective approach for the industrial processing of o/w emulsions because it is fast, simple, less energy-intensive, offers great flexibility for design scalability, avoids chemical additives, and reduces the oil concentration even below the acceptable regulatory discharge limit [9,10,11]. Therefore, much research is being done on the design and fabrication of o/w separation membranes [12,13]. Currently, polymeric membranes are the most widely used category of membranes due to their simple functionality, high porosity, and low cost. However, most polymeric membranes are relatively less hydrophilic and are, thus, easily prone to fouling [14]. In addition, their lack of physical and chemical stability makes them less competitive when used for extended periods of time under harsher conditions [15], requiring more robust alternatives, such as ceramic membranes [16,17,18].

Despite the superior chemical and mechanical properties of ceramic membranes over polymer membranes, fouling is an inevitable phenomenon in membrane separation processes. Membrane fouling in oily wastewater separation is primarily due to the deposition of oil droplets on the membrane’s surface. Depositions clog membrane pores and reduce membrane performance and longevity over time [19]. Therefore, it is important to implement a proper modification of ceramic membranes to obtain a membrane surface that is less prone to fouling. Few studies have revealed that membrane surface properties play a critical role in fouling mitigation [20,21]. For example, the rapid development of nanotechnology in the last few years has demonstrated the antifouling capabilities of nanometal oxides and their use in the surface modification of membranes. One of the most common nanometal oxide-inspired antifouling strategies, as far as the treatment of oily wastewater is concerned, is surface hydrophilization [22,23,24,25,26,27,28,29]. In general, hydrophilization increases the oleophobicity of membranes under water by increasing their hydrophilicity, thus minimizing contact between oil droplets and membrane surfaces. Nanometal oxides are intrinsically hydrophilic due to the presence of abundant surface hydroxyl groups (OH^−^) formed from the dissociative chemisorption of water molecules [30]. The selective affinity for water molecules caused by the OH^−^ groups hinders the formation of an oily cake layer on the membrane surface, leading to improved oil rejection and permeate flux [31].

In a study conducted by Chang et al. [26] to separate the o/w emulsion, commercial Al_2_O_3_ membranes were dip-coated with hydrophilic Al_2_O_3_ nanoparticles (NPs), resulting in a 20% increase in water flux compared to the unmodified membrane. Similarly, Zhou et al. [27] coated commercial Al_2_O_3_ membranes with ZrO_2_ NPs through in situ hydrolysis of ZrCl_2_. The steady flux of the modified membrane was at 88% of the initial flux (compared to 30% retainment of the initial flux for the unmodified membrane), and the oil rejection was at least 97.8%. In another study, Hu et al. [32] coated Al_2_O_3_ with graphene oxide through vacuum filtration and observed a 27.8% increase in flux with a higher oil rejection rate compared to the unmodified membrane. Zhang et al. [23] coated Al_2_O_3_ ceramic membranes with TiO_2_ nanorods by magnetic sputtering and hydrothermal reaction. The coating ensured superhydrophilicity and a high oil rejection of 99.1%. Apart from Al_2_O_3_ membranes, TiO_2_ and ZrO_2_ are other commonly used ceramic membranes for separating o/w emulsions. Lu et al. [22] coated TiO_2_/ZrO_2_ membranes with Fe_2_O_3_ through a pulse-layer deposition. The coated membranes exhibited better antifouling ability, thus increasing flux recovery by 10%. In a similar study, Lu et al. [29] determined the fouling tendency of ZrO_2_ membranes coated with different nanometal oxides, such as TiO_2_, Fe_2_O_3_, MnO_2_, CuO, and CeO_2_. The Fe_2_O_3_ coating was found to impart the highest hydrophilicity and, accordingly, the lowest incidence of fouling. Recently, Marzouk et al. [25] dip-coated the TiO_2_ membrane with SiO_2_ NPs, and the coated surface was found to be superhydrophilic with a TOC removal of 91%.

Among nanometal oxides, copper oxide (CuO) is one of the most important multifunctional semiconductor materials with excellent electrical, mechanical, photocatalytic, and chemical properties. Its excellent ability to adsorb -OH groups makes it superhydrophilic and underwater-superoleophobic [33]. Additionally, CuO has significant antimicrobial properties, so it can potentially suppress the formation of biofilms on the surface of the membrane, considering that the produced water may contain bacteria colonies [34,35]. The use of CuO nanoparticles for membrane-based water treatment has been extensively researched, as they enhance membrane adsorptive properties and their overall performance, thus extending the lifespan of the membranes. These nanoparticles improve surface hydrophilicity, regulate pore size/porosity, and exhibit a strong resistance against oil corrosion, making them suitable for treating oil-polluted wastewater. In addition, the comparatively low cost of CuO, due to its natural abundance, makes it a better alternative, thus meeting the demand for an effective, lower-cost hydrophilic material for the surface modification of membranes. Furthermore, studies have shown that silver functionalization of CuO (Ag-CuO) greatly enhances o/w separation with excellent antimicrobial efficiency compared to pristine CuO [36,37]. Doping of CuO with Ag improved the already existing hydrophilic (and antimicrobial) properties of the CuO NPs since heterostructured nanometal oxides are known to possess superior physical and chemical properties compared to the monocomponent metal/oxide structures [38,39].

Thus, the modification of commercial ceramic membranes with Ag-CuO NPs is expected to have desirous benefits, such as improved flux, better antifouling ability, and improved oil rejection for the treatment of oily wastewater. In this study, TiO_2_/ZrO_2_ ceramic composite membranes were coated with Ag-CuO NPs using the facile dip-coating method. The nanocomposite was intrinsically synthesized by a hydrothermal approach and systematically characterized. The effects of the Ag-CuO coatings on the physiochemical characterizations, o/w separation performance, and antifouling ability of the membranes were examined. In terms of novelty, this work addresses a simple and cost-effective surface modification of commercial TiO_2_/ZrO_2_ ceramic membranes that results in higher flux, improved oil rejection, and enhanced antifouling performance for oily wastewater treatment. Moreover, the membranes modified with Ag-CuO NPs demonstrated good recyclability, long-term stability, and good corrosion resistance for lab-scale investigations. Considering these specified features, the Ag-CuO decorated TiO_2_/ZrO_2_ ceramic membranes might be suitable for treating real-world oily wastewater streams.

## 2. Materials and Methods

### 2.1. Materials

Commercial TiO_2_/ZrO_2_ composite ceramic UF membranes (2.5 mm thickness, 1 kDa MWCO, and 47 mm diameter) were supplied by Sterlitech Corporation, Auburn, WA, USA. Copper (II) nitrate pentahydrate (Cu(NO_3_)_2_.5H_2_O, M_w_ = 249.7 g/mol), silver nitrate (AgNO_3_, M_w_ = 169.87 g/mol), sodium hydroxide (NaOH), ethylene glycol (C_2_H_6_O_2_, 62.07 g/mol), and polyvinylpyrrolidone (PVP, M_w_: 40,000 kDa) were procured from Merck, Bengaluru, India. Sodium dodecyl sulfate (SDS), purchased from Sigma-Aldrich, Berlin, Germany, was used for membrane cleaning. Deionized (DI) water (resistivity of 15 MΩ cm at 25 °C) obtained from a Milli-Q purification system (Millipore Corp., Burlington, MA, USA) was used for all experiments. All chemicals were of analytical grade and used without any further purification.

### 2.2. Methods

#### 2.2.1. Preparation and Characterization of Ag-CuO NPs

A facile hydrothermal process was used to synthesize Ag-CuO NPs. About 0.15 g of Cu(NO_3_)_2_.5H_2_O was dissolved in 50 mL DI water and stirred for 10 min. Subsequently, 0.7 g of ethylene glycol and 0.1 g AgNO_3_ were added to the solution and stirred until complete dissolution. Furthermore, a 5 M solution of NaOH was added slowly under constant vigorous stirring at 70 °C until a pH of 11 was achieved. The mixture was then ultrasonicated for 20 min, transferred to a Teflon autoclave, and oven-heated at 170 °C for 24 h. The precipitate obtained was washed several times in DI water and acetone to remove any impurities. The final filtered product was dried at 80 °C for 12 h and subsequently calcinated at 300 °C for 4 h to obtain the Ag-CuO NPs. The crystal structure of the synthesized nanoparticles was analyzed using an X-ray diffractometer D2 Phaser (Bruker, Berlin, Germany) with Cu-K_α_ radiation, wavelength 1.541 Å, and operating voltage 30 kV. The structure, shape, and size of the Ag-CuO NPs were studied under a scanning electron microscope, SEM JSM 7610F (JEOL, Tokyo, Japan), and its chemical composition was examined using an Energy Dispersive Spectrum (Oxford Instruments, Tokyo, Japan) connected to the scanning electron microscope.

#### 2.2.2. Preparation and Characterization of Modified Ceramic Membranes

Three membranes labeled M0.1, M0.5, and M1.0 were dipped in respective Ag-CuO NPs suspensions of concentrations 0.1 wt.%, 0.5 wt.%, and 1.0 wt.% for 30 min. The NPs suspensions were stabilized with 5 wt.% PVP. The coated membranes were removed from the suspensions and dried in a hot air oven at 70 °C for 12 h, followed by calcination at 300 °C for 3 h. The hybrid membranes were then allowed to cool for 60 min and stored for subsequent characterizations and experiments. A pristine unmodified membrane (M0) was used for the control experiments.

The open porosity of the membrane was measured using DI water and calculated using Archimedes’ principle [40]. The dry weights of the membranes were first determined as *W_1_*. The membranes were thereafter soaked in DI water for 24 h. The wet weights, W_2_, were measured after removing the membranes from the DI water and superficially wiping the surface. The membrane porosity, *ε*, was determined using Equation (1) [41].
(1)$$\epsilon =\frac{W_{2}-W_{1}}{W_{1}}\times 100$$

All the prepared membranes were further characterized for morphology, chemical composition, and surface wettability. Scanning electron microscopy (JSM 7610F, JEOL, Tokyo, Japan) was used to determine the morphological features of the coated membrane surface. Energy-dispersive x-ray spectroscopy (EDX) at an electron-accelerating voltage of 15 kV was used to confirm the elemental composition of the coated surfaces. The hydrophilicity of the coated membrane was determined by contact angle measurements via the sessile drop method (0.3 μL dispense volume of DI water) using a Kyowa CA-A DM-501 (Kyowa Ltd., Tokyo, Japan) device. Contact angle measurements were performed in triplicates at different spots of the membranes, and the average value was reported.

#### 2.2.3. Preparation and Characterization of O/W Emulsion

To investigate the oil removal performance of the coated membranes, 1 g/L o/w emulsion was prepared by adding 1 g gasoline (Special-95 grade purchased at a local gasoline station in Abu Dhabi) and 0.5 wt.% Tween-20 to 1 L of DI water. The mixture was stirred at 3000 rpm for 3 h and further ultrasonicated at 40 kHz for 1 h at room temperature. The o/w emulsion was characterized for the oil-droplet size distribution using an optical microscope (Optika B-159) equipped with a digital camera (C-B5, Optika, Milan, Italy) and ImageJ software for the droplet size analysis. Other physicochemical properties, namely viscosity, density, and pH of the o/w emulsion, were also determined.

#### 2.2.4. Permeation and Oil Rejection Performance of Membranes

The dead-end vacuum filtration method was used to determine both the pure water flux (*J_w1_*) and the o/w permeate flux (*J_f_*) of each membrane at a transmembrane pressure of 1 bar. The pump model used was WELCH 2546C-02A (Gardner Denver Thomas, Inc., Sheboygan, WI, USA). For *J_w1_*, a graduated Buchner funnel was filled with 2000 mL of DI water, and the average time required to collect 1000 mL of permeate (10 batches of 100 mL permeate collection) was observed. The same protocol was followed to determine *J_f_*, except that DI water was replaced with the synthetic o/w emulsion. With an effective filtration area of 13.2 cm^2^, the permeate flux was calculated using Equation (2).
(2)$$J=\frac{V}{A\times \Delta t}$$
where *V* (L) is the volume of permeate, *A* (m^2^) is the effective surface area of the membrane, and Δ*t* (h) is the filtration time.

To determine the oil removal efficiency of the control and hybrid membranes, the permeate concentrations were determined by UV-vis spectrometry, Hack Lange DR 5000, (Hach, Loveland, CO, USA). First, the characteristic wavelength of absorbance (270 nm) for the permeate was identified through a full-scan spectrum in the region of 190 to 800 nm. A calibration curve was then prepared by matching peak absorbances at 270 nm with o/w emulsions of known concentrations. The oil removal capacity (*R*) of the membrane was subsequently determined using Equation (3).
(3)$$R=\frac{C_{f}-C_{p}}{C_{f}}\times 100$$
where *C_f_* and *C_p_* are the feed and permeate concentrations, respectively, in mg/L.

#### 2.2.5. Antifouling Studies

The antifouling properties of both the pristine and coated membranes were preliminarily investigated to select the optimal-performance membrane for further studies. After obtaining *J_w_*_1_ and *J_f_* from the filtration step in 2.2.4, the membranes were hydrodynamically cleaned by backwashing with 1 wt.% SDS solution and DI water at a fixed pressure of 2 bar. Then, for a second time, pure water fluxes (*J_w_*_2_) were recorded for each of the cleaned membranes. Flux recovery ratio (FRR), as well as total flux decline ratio (*R_t_*), reversible flux decline ratio (*R_r_*), and irreversible flux decline ratio (*R_ir_*), were calculated from Equations (4)–(7), respectively [42]. To investigate the long-term usability of the membranes, the membrane with the optimal antifouling performance, together with the unmodified membrane as control, was subjected to multiple cycles of o/w filtration, and changes in flux values over the reruns were noted. Additionally, the continuous filtration stability of the optimal membrane was examined by carrying out the o/w emulsion filtration for 300 min without any intermittent membrane washing.
(4)$$FRR\left(\%\right)=\frac{Jw_{2}}{Jw_{1}}\times 100$$
(5)$$R_{t}\left(\%\right)=\left(\frac{J_{w1}-J_{f}}{J_{w1}}\right)\times 100$$
(6)$$R_{r}\left(\%\right)=\left(\frac{J_{w2}-J_{f}}{J_{w1}}\right)\times 100$$
(7)$$R_{ir}\left(\%\right)=\left(\frac{J_{w1}-J_{w2}}{J_{w1}}\right)\times 100$$

#### 2.2.6. Corrosion-Resistance Test

The corrosion resistance of the optimal performing membrane was examined by immersing it separately in hydrochloric acid (HCl, pH 2) and sodium hydroxide (NaOH, pH 12) solution for 72 h [40]. Moreover, the chemical stability of the membrane against actual seawater (pH 7.9) and o/w emulsion (at a higher concentration of 2 g/L, pH 5.6) was also investigated by the same procedure to understand the suitability of the membrane for real-world situations. Membrane stability against corrosion was measured by determining their weight loss before and after the corrosion test. Further, the morphology and elemental composition of the membranes, pre-, and post-corrosion analysis, were investigated through SEM and EDX techniques. Furthermore, changes in surface composition were studied through Fourier Transform Infrared (FT-IR) spectroscopy, using Vertex 80v FT-IR spectrometer (Bruker, Berlin, Germany). The wavenumbers ranged from 4000 cm^−1^ to 400 cm^−1^, and the resolution was 4 cm^−1^.

## 3. Results and Discussion

### 3.1. Ag-CuO NPs Characteristics

Figure 1A reveals the morphology of the Ag-CuO NPs as a rod-like structure predominantly. There appeared to be a slight degree of agglomeration that can be attributed to the high surface energy of the NPs. The average particle size of Ag-CuO NPs was determined to be 75.2 nm. Figure 1B shows the EDX spectrum of the Ag-CuO composite that confirmed the elemental composition of the synthesized NPs. The low peak of Ag is due to its relatively low concentration in the mixture, while the Au and Pd peaks are attributed to the sputtering process conducted to improve the SEM imaging. Figure 2 shows the XRD spectra of the nanocomposites with sharp and well-defined peaks, which signifies the high crystallinity of the Ag-CuO NPs. The 2θ peaks occurring at 38.20°, 44.4°, 64.44°, and 77.40° matched with the standard JCPDS card #04-0783 (for Ag) and can be indexed to the planes (111), (200), (220), and (311) of the face-centered cubic structure of Ag phase, respectively. Other characteristic diffraction peaks (aside from Ag) matched well with the standard JCPDS card #048-1548 for the monoclinic CuO phase. The absence of any other peak confirmed the product purity of the synthesized Ag-CuO NPs. Thus, the morphology and structural studies of the Ag-CuO particles established the nanoscale dimension, crystalline nature, and product purity of the hydrothermally synthesized nanocomposite.

### 3.2. Characteristics of Ag-CuO-Coated Hybrid Membranes

Figure 3 shows the visual representations of the pristine and modified ceramic membranes, clearly differentiated by a change in the surface color due to the incorporation of Ag-CuO. With increasing levels of the NP coatings, the membrane color changed from white to brown. The color distribution was also uniform, which indicated the uniform coating of the NPs on the membrane surface. Figure 4A shows the SEM micrographs of the top surface of the membranes. The surface appeared smooth for the M0 membrane, while the coated membranes showed adequate accumulation of particulate matter dispersed on the entire membrane surface. This indicated a plausible increase in the surface roughness of the membranes due to the interaction with the NPs [21]. According to Cassie’s model, a critical factor for super-wetting behavior is expanded roughness [43]. Thus, the analysis confirmed the presence of the Ag-CuO NPs on the coated membrane surfaces. Furthermore, a comparison of the membrane cross-section for M0 and M1.0 (Figure 4B) established the particulate nature of the NPs’ attachment to the membrane surfaces, as no continuous secondary filtration layer was observed. Furthermore, EDX images of coated membranes (Figure 5) confirmed the presence of Ag-CuO NPs on the membranes’ surfaces, revealing the elemental compositions of both the membranes and the nanocoating.

The surface hydrophilicity of the membranes was quantified through contact angle measurements, and the results for the same are shown in Figure 6A. In general, low contact angles are desired to facilitate the transport of water molecules across the membrane, as high contact angles (>90°) indicate an oleophilic surface. A contact angle of 56° for the M0 suggested that the pristine membrane was fairly hydrophilic. The attachment of Ag-CuO NPs to the surfaces of the membranes further enhanced their hydrophilicity, and the contact angle systematically dropped to 36° for M1.0. The largest drop in the contact angle of 12° was between M0.1 (51°) and M0.5 (39°). However, the decrease in contact angle from M0.5 (39°) to M1.0 (36°) was only 3°. The excessive surface loading of M1.0 likely caused a reduction in the membrane pore size that affected its surface wettability, accounting for the marginal difference observed between it and M0.5 [44,45]. As the ceramic membranes are designed for o/w separation applications, it is expected that the modified ceramic membrane must be highly hydrophilic and oleophobic for efficient oil rejection. Further, Figure 6A shows the porosity of the prepared membranes in relation to Ag-CuO NPs concentration. Results showed that the porosity decreased from 17.3 to 8.7% with an increase in Ag-CuO NPs concentration from 0 to 1 wt.%. Membrane porosity decreased slightly from 0 to 0.5 wt.% of NPs concentration but sharply from 0.5 wt.% to 1 wt.%. The latter effect was mainly due to the agglomeration of the NPs leading to pore constriction and blockage. The results were in good agreement with the SEM studies. Thus, the surface modification of the commercial TiO_2_/ZrO_2_ ceramic membranes with Ag-CuO NPs enhanced the surface hydrophilicity and lowered the membrane porosity owing to the uniform distribution of the nanocomposites.

### 3.3. Pure Water Flux Test and Membrane Performance in O/W Removal

The o/w emulsion was considered highly stable, as there was no coalescence of oil droplets on the surface even after storage for weeks. Figure 6B represents the optical micrograph of the o/w emulsion that revealed almost spherical oil droplets uniformly dispersed in a continuous water phase. The average oil droplet size was determined as 1.82 ± 0.18 µm. The pH of the emulsion was 6.8, while its density and kinematic viscosity were 0.99748 g/cm^3^ and 0.9020 mm^2^/s, respectively, at 25 °C.

The results for the pure water flux are presented in Figure 7A. It was observed that all of the modified membranes showed a flux higher than the control. This elucidated the role of permeation enhancement of the Ag-CuO NPs due to their strong hydrophilic nature. Furthermore, there was an increase in flux with an increase in Ag-CuO loading up to 0.5 wt.%. An increase in the nanocomposite concentration led to pronounced hydrophilic effects in the coated membranes (as evidenced by contact angle measurements). This ultimately enhanced the pure water flux by ~30% for M0.5 as compared to the control M0 membrane. Similar results for flux improvements have been reported by Marzouk et al. [25] on SiO_2_-coated TiO_2_ membranes. However, the flux decreased significantly for M1.0 in relation to M0.5. The seeming anomaly could be explained in terms of the interplay of the diffusive transport mechanism and the convective bulk flow that took place. In general, improved hydrophilicity enhances diffusion through the molecular interaction of water and the membrane matrix, while improved porosity augments solvent transport via convective bulk flow [46,47,48]. While the Ag-CuO NP coating improved the surface hydrophilicity of the membranes, it also impaired the membrane porosity. This was evident from the morphology and porosity studies of the membranes. Specifically, the M1.0 membrane had a very low porosity of 4.6%, which indicated that the membrane was only 50% porous in comparison to the M0 membrane. Thus, it is suggested that diffusive transport owing to membrane hydrophilicity took precedence without affecting convective transport up to 0.5 wt.% loading, thus increasing the pure water flux through the membranes. On the other hand, higher loading rates of the NPs led to a continuous reduction in membrane porosity leading to a substantial loss of convective transport, which resulted in relatively low water flux. Such observations are consistent with the results of similar studies conducted by Li et al. [49] and Marzouk et al. [25].

Figure 7B shows the o/w permeate fluxes that were similar to the pure water flux pattern; however, they were lower than the corresponding water fluxes due to the deposition of oil droplets. When it came to oil rejection, it was observed that rejection improved slightly with the Ag-CuO NPs loading in the membranes (Figure 7B). The abundant OH^−^ groups of the metal oxide nanoparticles enhanced the hydrophilicity and oleophobicity of the surface, leading to reduced interactions between the membrane surface and the oil droplets [23]. As a result, the oil rejection performance was relatively higher for all coated membranes than the pristine membrane. In addition, it was observed that there was an increase in oil rejection with an increase in NP loading due to the pronounced oil repulsion nature of the nanocomposite. Moreover, the progressive reduction in membrane porosity (Figure 6A) with the incremental modifier concentration contributed to oil rejection [40]. High oil rejection efficiencies of 97.8% and 98.6% were observed for the M0.5 and M1.0 membranes. Thus, the ultrafiltration analysis of the membranes showed that all the modified membranes displayed higher flux and better oil rejection performance than the unmodified membrane. Specifically, the membrane incorporated with 0.5 wt.% of Ag-CuO NPs showed the highest flux and oil rejection characteristics among the modified membranes.

### 3.4. Antifouling Ability and Reusability Study Results

A fundamental problem affecting water flow and separation efficiency is the fouling or clogging of membranes by oil droplets. Consequently, antifouling membranes are essential for the treatment of oily wastewater. The results of the antifouling studies for the membranes examined in this work are presented in Table 1. From the initial assessment of the antifouling performance of the membranes, it was clear that the Ag-CuO NP coating improved the overall membrane resistance to oil fouling, which was evident from their higher FRR values than the uncoated ones. Within the modified membranes, the FRR increased with increasing concentration of the NPs, which indicated the antifouling ability of the nanocoatings and their strong interaction with the membrane surface.

In general, flux decline is normally explained by two main components: reversible and irreversible fouling effects. While both reversible and irreversible fouling hamper membrane performance, the former can easily be “reversed” through simple hydrodynamic cleaning methods, while the latter usually requires extensive chemical cleanup or complete membrane replacement. Thus, irreversible fouling is the main cause of membrane performance decay and loss of longevity [50]. The results in Table 1 show that M0 recorded not only the highest reversible and irreversible flux decline ratios but also that the irreversible flux decline ratio was higher than the reversible flux decline ratio. However, with the addition of the Ag-CuO NPs, both reversible and irreversible flux decline ratios progressively reduced, and the reversible decline ratio was higher than the irreversible flux decline ratio for all of the modified membranes. This was a notable achievement for the antifouling performance of the Ag-CuO NP-coated ceramic membranes. The total flux decline ratio was reduced from 43.98% for the M0 control to 22.55% for the M1.0 composite membrane.

Close observations of the results showed that, even though M1.0 had the highest flux recovery and appeared to be the least fouled, the performance of M0.5 in this regard was equally comparable. Furthermore, M0.5 recorded the highest pure water flux, which was 8% higher than M1.0. Thus, M0.5 was considered the optimal performing membrane and was further subjected to the recyclability test, with M0 serving as the control. In general, the recyclability test demonstrated the stability and robustness of the membrane to withstand long periods of operation. Thus, the selected membranes were subjected to five cycles of o/w emulsion filtration, and each cycle was carried out for 60 min with subsequent membrane cleaning using 1 wt.% SDS solution and DI water. Results for the reusability studies are shown in Figure 8A. At the end of the fifth cycle, it was observed that there was only a 6% drop in flux for M0.5 compared to a 15% drop for M0. Although the cleaning method may not have fully recovered the membrane surface, the Ag-CuO NP coating ensured a better antifouling performance of the M0.5 membrane, as evidenced by its low flux loss compared to the unmodified M0 membrane. Further, the results for the long-term filtration run of o/w emulsion separation by the modified and unmodified membranes are depicted in Figure 8B. A continuous o/w separation run for 300 min showed a more stable and better performance of the coated membrane (M0.5) than the uncoated membrane (M0). The modified membrane reached a higher stable flux in a shorter time (50 min) than the pristine membrane (160 min). Results elucidated a better antifouling tendency, reusability, and operational stability of the Ag-CuO NP-coated membrane than the pure membrane.

### 3.5. Corrosion Resistance

The corrosion resistance of the optimal M0.5 membrane was tested in different environments to realize the practical application of the membrane for real-world situations. Experiments on simulated acid and alkali medium, as well as actual seawater and o/w emulsion, were performed. Images of the corrosion-resistance test of the M0.5 membrane against seawater and o/w emulsion are shown in Figure 9, and the results for the corrosion-resistance studies are presented in Figure 10. The visual images (Figure 10A–C) of the fresh and soaked M0.5 membranes (72 h in seawater and o/w emulsion) showed no distinctive variations. SEM imaging of the membranes (Figure 10D–F) confirmed the intact morphology of the soaked membranes, which was evident from the absence of any fractures, cracks, or pores on the surface [51]. Further, the chemical stability of the membrane was evaluated by calculating the weight loss after soaking the sample in different solutions. As shown in Figure 10G, it can be concluded that the membrane exhibited excellent chemical stability with a trivial weight loss of ≤1% in harsh conditions. Notably, the membrane recorded a negligible weight loss of 0.3% and 0.75% in o/w emulsion and seawater, respectively. Furthermore, FT-IR studies (Figure 10H) revealed that the surface chemistry of the membranes was unaltered, with good resistance to corrosion against seawater and o/w emulsion. Figure 10I shows the elemental composition of the M0.5 membrane before and after corrosion testing obtained through EDX studies. EDX results showed that the elemental membrane composition was identical before and after corrosion tests. The results were in good agreement with the SEM, weight loss, and FT-IR studies of the membranes. Concerning the ecotoxicity of the CuO nanoparticles, the amount of CuO used in this study for the membrane modification was very low (0.1–1.0 wt.%), which ensures the safety of treating oil-polluted wastewater. Likewise, as shown in Figure 10I, the elemental analysis of the M0.5 membrane (pre- and post-treatment scenarios) confirmed the non-leaching of the nanoparticles from the membrane matrix, thus guaranteeing the nontoxicity of the treated water. Based on corrosion studies, it can be concluded that the optimal Ag-CuO-coated membrane possessed excellent corrosion resistance, making it suitable for actual o/w separation.

## 4. Conclusions

This work demonstrated for the first time the efficient separation of o/w emulsions via Ag-CuO NP-modified ceramic membranes. Rod-like Ag-CuO NPs with high crystallinity and product purity were prepared hydrothermally and applied to modify commercial T_i_O_2_/ZrO_2_ ceramic UF membranes using a simple dip-coating technique. Membranes modified with Ag-CuO NPs showed improved filtration characteristics and o/w separation as follows:Higher hydrophilicity and lower porosity with incremental levels of the NPs;Pure water flux was enhanced by a maximum of 30% compared to unmodified membrane;Oil rejection improved from 89.4% to 98.6% due to Ag-CuO NPs loading;Increased flux recovery ratio (~15%) and reduced irreversible fouling (~14%) compared to unmodified membrane;Membrane, modified with 0.5 wt.% NPs displayed excellent o/w filtration characteristics, good reusability, operational stability, and corrosion resistance.

Thus, the straightforward and simpler surface modification approach, the resulting enhanced hydrophilicity/oleophobicity, the higher flux, and oil rejection rates with good stability after cyclic operation prove the effectiveness of Ag-CuO NP-coated membranes for efficient treatment of o/w emulsions.

## Figures and Tables

**Figure 1 membranes-13-00176-f001:**
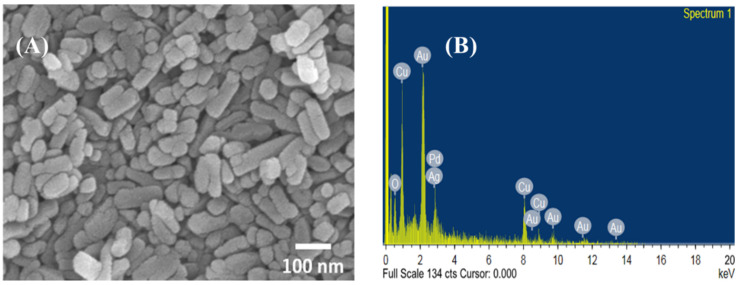
(**A**) SEM micrograph and (**B**) EDX spectrum of Ag-CuO NPs.

**Figure 2 membranes-13-00176-f002:**
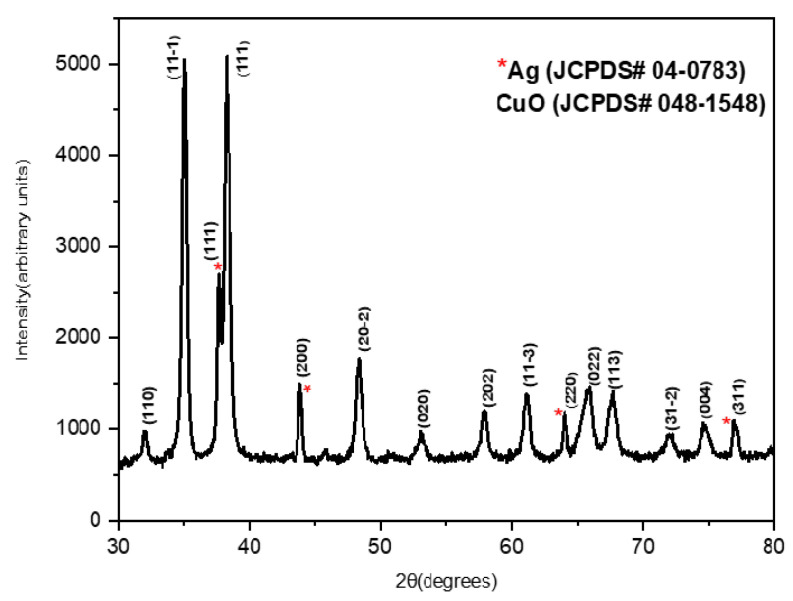
XRD spectrum of Ag-CuO NPs (* represents Ag peaks).

**Figure 3 membranes-13-00176-f003:**
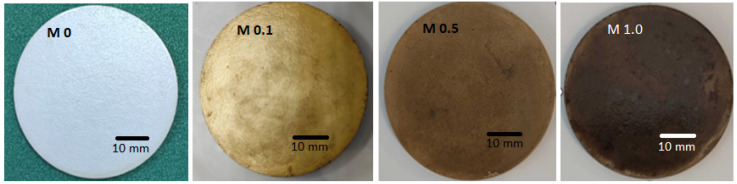
Optical images of pristine and Ag-CuO-coated membranes.

**Figure 4 membranes-13-00176-f004:**
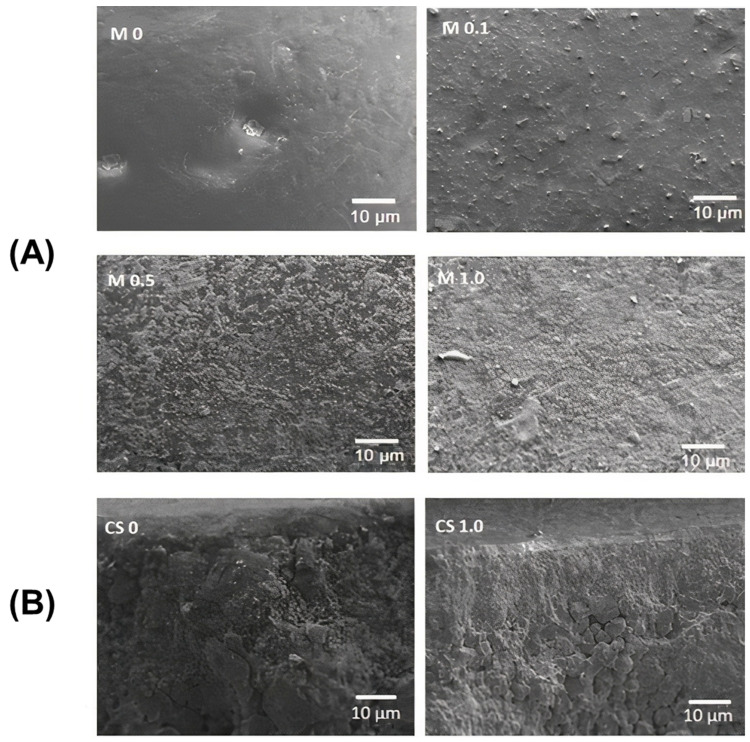
SEM images of (**A**) top surface and (**B**) cross-section of the membranes.

**Figure 5 membranes-13-00176-f005:**
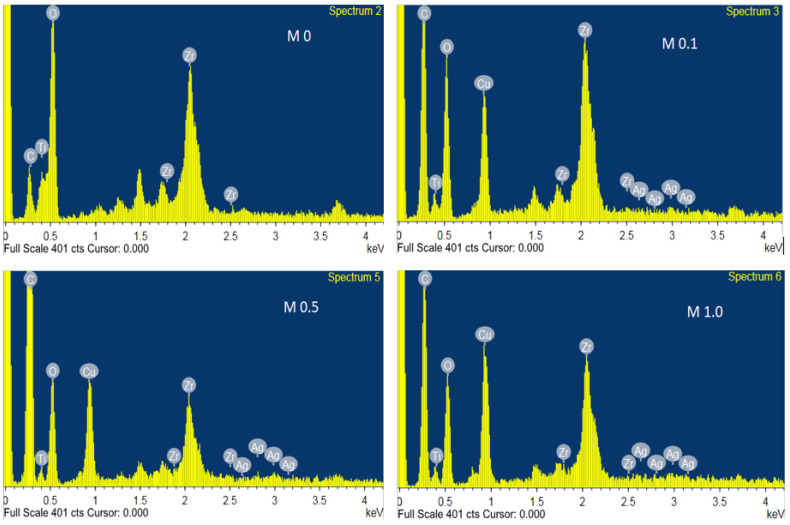
EDX spectra of pristine and coated membranes.

**Figure 6 membranes-13-00176-f006:**
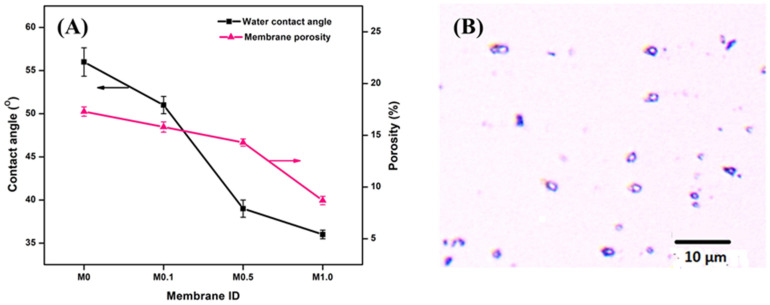
(**A**) Contact angles and porosity of pristine and coated membranes and (**B**) Optical microscopy image of o/w emulsion.

**Figure 7 membranes-13-00176-f007:**
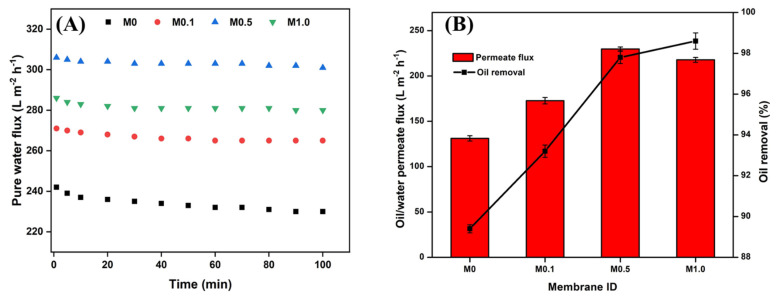
(**A**) Pure water fluxes and (**B**) o/w separation performance of pristine and coated membranes.

**Figure 8 membranes-13-00176-f008:**
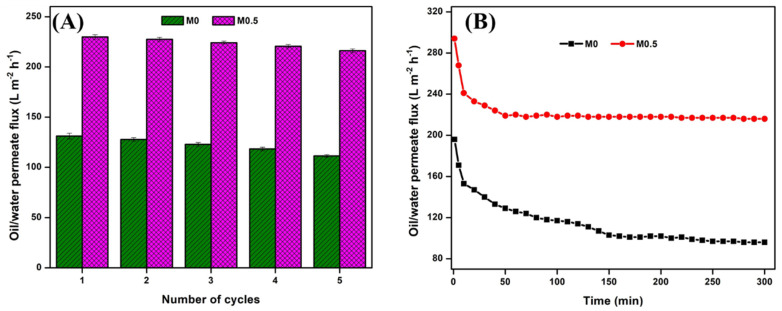
(**A**) Cyclic stability and (**B**) Long-term o/w filtration performance of M0 (control) and M0.5 membranes.

**Figure 9 membranes-13-00176-f009:**
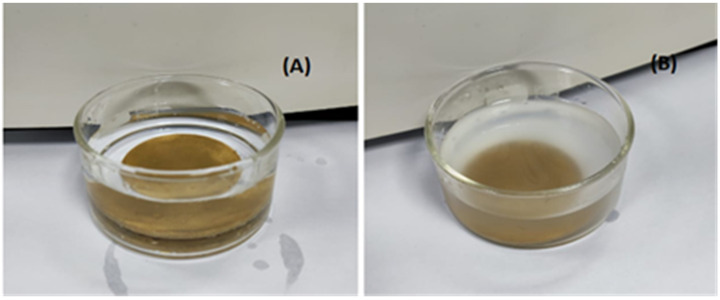
(**A**) M0.5 immersed in seawater (**B**) M0.5 immersed in o/w emulsion with higher oil concentration of 2 g/L.

**Figure 10 membranes-13-00176-f010:**
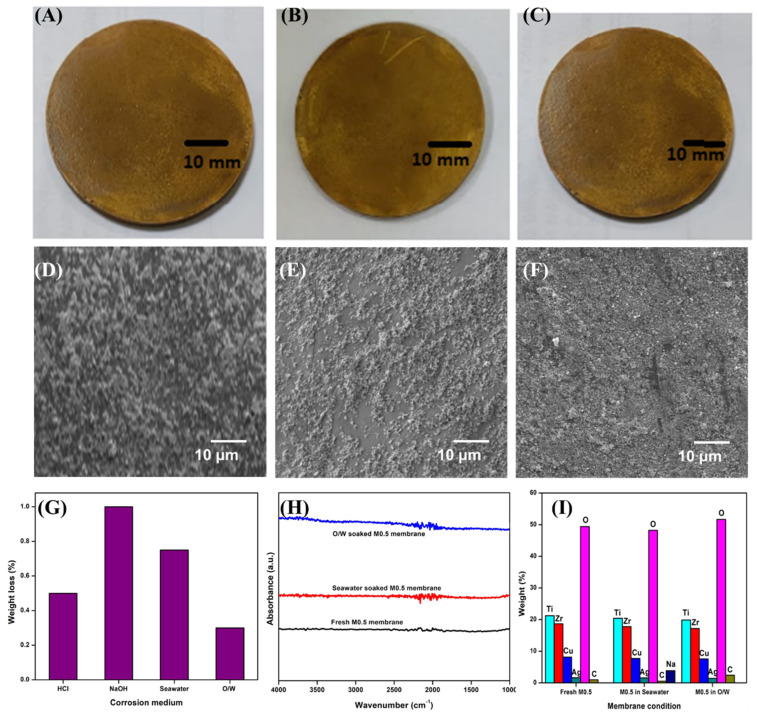
(**A**) Fresh M0.5 membrane (**B**) M0.5 after immersion in seawater for 72 h (**C**) M0.5 after immersion in o/w emulsion for 72 h; (**D**–**F**) SEM micrographs of fresh-seawater-soaked and o/w-emulsion-soaked M0.5 membrane; (**G**) Weight loss of M0.5 in different corrosion medium; (**H**) FT-IR spectra of the M0.5 membrane; (**I**) Chemical composition of M0.5 membrane.

**Table 1 membranes-13-00176-t001:** Antifouling performance of the pristine and Ag-CuO NP-coated membranes.

Membrane ID	*J_w_*_1_ (L m^−2^ h^−1^)	*J_f_* (L m^−2^ h^−1^)	*J_w_*_2_ (L m^−2^ h^−1^)	FRR (%)	*R_r_* (%)	*R_ir_* (%)	*R_t_* (%)
M 0	234.12	131.15	180.71	77.19%	21.17%	22.81%	43.98%
M 0.1	266.32	172.68	221.39	83.13%	18.29%	16.87%	35.16%
M 0.5	303.63	229.83	275.15	90.62%	14.93%	9.38%	24.31%
M 1.0	281.19	217.79	257.59	91.61%	14.15%	8.39%	22.55%

## Data Availability

The data presented in this study are available on request from the corresponding author.

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
