# Peer review of "Ag-CuO-Decorated Ceramic Membranes for Effective Treatment of Oily Wastewater"

_membranes, 2023, doi:10.3390/membranes13020176_

Round 1

Reviewer 1 Report (Previous Reviewer 1)

The paper entitled ''Ag-CuO Decorated Ceramic Membranes for Effective Treatment of Oily Wastewater'' aims to investigate Ag-CuO nanoparticles as surface modifiers to improve the performance of ceramic membranes for the treatment of oily wastewater. The study is interesting. A comprehensive analysis of membrane fouling was reported. The manuscript is well-written. This manuscript can be considered for publication in Membranes after a revision. My comments are as follows.

1.     Abstract, Line 16: What percentage of improvement was achieved in the flux with the 0.5 wt.% modified membrane?

2.     Results and discussion: Please give the permeate flux data with time. What is the duration of each experiment?

3.     Results and discussion, Figure 2: Please make the titles in the figure legible.

4.     Results and discussion, Figure 6b: The quality of the microscopy image is not good. Please replace it with a new one.

5.     Results and discussion, Figure 7b: I noticed that the M0 membrane also has a high oil rejection rate, >94%. I think the improvement in the oil rejection for the modified membranes is negligible. So, it does not make sense to highlight this in the paper. Please also discuss this result in the manuscript. In that case, the improvement was observed only in the permeate flux.

6.     Results and discussion, Figure 7a: Why is the pure water flux of the M1 membrane lower than that of the M0.5 membrane?

7.     Results and discussion: Please also give flux data with time for the reusability test.  

8.     Figure 5: Please discuss more of the elemental compositions of the membranes before and after filtration experiments.

9.     Add a summarizing sentence on what should be remembered after each section in results and discussion.

10.  Conclusions: This section should be written by evaluating interesting results, and suggestions for future applications should be given. The format that concise text followed by bullets is recommended for the conclusions section.

Author Response

The authors are very much thankful to Reviewer-1 for the thorough review of the manuscript and for providing valuable comments to improve the quality of the revised manuscript. Necessary changes have been made to the manuscript according to the suggestions and recommendations of the reviewer. Detailed responses to the review comments of Reviewer-1 are presented in the attachment. The authors also express their immense gratitude for recommending the manuscript for publication.

Reviewer 2 Report (New Reviewer)

Author Response

The authors are very much thankful to Reviewer-2 for the thorough review of the manuscript and for providing valuable comments to improve the quality of the revised manuscript. Necessary changes have been made to the manuscript according to the suggestions and recommendations of the reviewer. Detailed responses to the review comments of Reviewer-2 are presented in the attachment. The authors also express their immense gratitude for recommending the manuscript for publication.

Reviewer 3 Report (Previous Reviewer 3)

All comments are marked in the attached file

Author Response

The authors are very much thankful to Reviewer-3 for the thorough review of the manuscript and for providing valuable comments to improve the quality of the revised manuscript. Necessary changes have been made to the manuscript according to the suggestions and recommendations of the reviewer. Detailed responses to the review comments of Reviewer-3 are presented in the attachment. The authors also express their immense gratitude for recommending the manuscript for publication.

Reviewer 4 Report (New Reviewer)

The review of Ag-CuO Decorated Ceramic Membranes for Effective Treatment of Oily Wastewater.

I find this paper a good one. I have no remarks about it without one important issue. As it has been proved CuO nanoparticles are one of the most toxic particles. The Authors should discuss this issue - is it safe to use such membranes for water filtration?

Author Response

The authors are very much thankful to Reviewer-4 for the thorough review of the manuscript and for providing valuable comments to improve the quality of the revised manuscript. Necessary changes have been made to the manuscript according to the suggestions and recommendations of the reviewer. Detailed responses to the review comments of Reviewer-4 are presented in the attachment. The authors also express their immense gratitude for recommending the manuscript for publication.

Round 2

Reviewer 1 Report (Previous Reviewer 1)

 I have carefully reviewed the R1 version of the manuscript. The authors have responded to my comments by addressing my major concerns and have improved the manuscript accordingly. I have no comment at this stage.

This manuscript is a resubmission of an earlier submission. The following is a list of the peer review reports and author responses from that submission.

Round 1

Reviewer 1 Report

The paper entitled ''High Flux Ag-CuO Decorated Ceramic Membranes for Effective Treatment of Oily Wastewater'' aims to investigate the performance of modified ceramic membranes with Ag-CuO nanoparticles for the oil contained wastewater. Implementing an effective method for the treatment of produced water is necessary for sustainable water management. The authors proposed using silver-functionalized copper oxide nanoparticles to modify ceramic membranes. They achieved high fluxes with the modified membranes. In addition, the modified membranes showed lower fouling ratios in irreversible fouling. I believe this paper can be accepted after major revision. My comments are as follows.

1.     Title: Please rewrite the title considering the key message of the paper. Remove ''high flux'' from the title.

2.     Abstract, The first sentence is too general. Please replace it with another one stating more specifically the problem addressed within the study.

3.     Abstract, Line 14: ''The as-prepared'' this is not clear to me. Is there a typo? Please check.

4.     Abstract, Lines 14-15: Please try to rewrite this sentence.

5.     Introduction, Lines 52-53: I do not think most polymeric membranes are hydrophobic. This statement should be revised.

6.     Introduction: In the introduction, membrane fouling is discussed and its importance is highlighted. However, I could not find a detailed discussion in the results and discussion section on membrane fouling.

7.     Introduction, Lines 110-111: Please mention the benefits expected.

8.     Introduction: Please underline the novelty of the study.

9.     Materials and Methods, Line 165: ''The as-coated'' is not clear to me. Please check.

10.  Materials and Methods, Section 2.2.3: The physicochemical properties of feed water should be given.

11.  The flux improvement with M05 membrane looks too high. I think the pure water flux experiment with this membrane should be repeated to make sure.

12.  50 mL filtered volume is too low to evaluate membrane fouling. At least 1 liter should be filtered. Please try to add additional experiments with large-volume filtration.

13.  Add a summarizing sentence on what should be remembered after each section in the results section.

14.  Conclusions: This section should be written by evaluating interesting results. The format that concise text followed by bullets is recommended to use.

Reviewer 2 Report

"To investigate the oil removal performance of the as-coated membranes, 1 g/L o/w 165 emulsion was prepared by adding 1 g gasoline (Special-95 grade purchased at a local gas-166 oline station in Abu Dhabi) and 0.5 wt. % Tween-20 to 1 L of DI water."

Wastewaters and Sea water is aggressive - it is necessary to check the corrosiveness.
Therefore, it is necessary to check whether the new material does not dissolve, e.g. in sew water. Without this, the work is incomplete and cannot be published.

Reviewer 3 Report

All notes and comments are marked in the attached file with the text of the paper.
